# Moderate confirmation bias enhances decision-making in groups of reinforcement-learning agents

**Clémence Bergerot**[1,2]*, **Wolfram Barfuss**[3,4], **Pawel Romanczuk**[1,5]

**1** Department of Biology, Humboldt Universität zu Berlin, Berlin, Germany, **2** Charité – Universitätsmedizin Berlin, Einstein Center for Neurosciences Berlin, Berlin, Germany, **3** Transdisciplinary Research Area: Sustainable Futures, University of Bonn, Bonn, Germany, **4** Center for Development Research (ZEF), University of Bonn, Bonn, Germany, **5** Science of Intelligence, Research Cluster of Excellence, Berlin, Germany

* clemence.bergerot@charite.de

**Data Availability Statement:** All relevant data are on figshare, DOI: 10.6084/m9.figshare.24916767 This data can be accessed via the following link: https://figshare.com/articles/dataset/Moderate_

## Abstract

Humans tend to give more weight to information confirming their beliefs than to information that disconfirms them. Nevertheless, this apparent irrationality has been shown to improve individual decision-making under uncertainty. However, little is known about this bias' impact on decision-making in a social context. Here, we investigate the conditions under which confirmation bias is beneficial or detrimental to decision-making under social influence. To do so, we develop a Collective Asymmetric Reinforcement Learning (CARL) model in which artificial agents observe others' actions and rewards, and update this information asymmetrically. We use agent-based simulations to study how confirmation bias affects collective performance on a two-armed bandit task, and how resource scarcity, group size and bias strength modulate this effect. We find that a confirmation bias benefits group learning across a wide range of resource-scarcity conditions. Moreover, we discover that, past a critical bias strength, resource abundance favors the emergence of two different performance regimes, one of which is suboptimal. In addition, we find that this regime bifurcation comes with polarization in small groups of agents. Overall, our results suggest the existence of an optimal, moderate level of confirmation bias for decision-making in a social context.

## Author summary

When we give more weight to information that confirms our existing beliefs, it typically has a negative impact on learning and decision-making. However, our study shows that a moderate confirmation bias can actually improve decision-making when multiple reinforcement learning agents learn together in a social context. This finding has important implications for policymakers who engage in fighting against societal polarization and the spreading of misinformation. It can also inspire the development of artificial, distributed learning algorithms. Based on our research, we recommend not directly targeting

confirmation_bias_enhances_collective_decision-making_in_reinforcement-learning_agents/24916767 Code can be found on the following GitHub repository: https://github.com/clembergerot/CARL.

**Funding:** This work was supported by funding under the form of a PhD scholarship by the Einstein Center for Neurosciences Berlin, Charité – Universitätsmedizin Berlin to CB and funding by the German Research Foundation under Germany's Excellence Strategy EXC 2002/1 "Science of Intelligence" project 390523135 to PR. The funders had no role in study design, data collection and analysis, decision to publish, or preparation of the manuscript.

**Competing interests:** The authors have declared that no competing interests exist.

confirmation bias but instead focusing on its underlying factors, such as group size, individual incentives, and the interactions between bias and the environment (such as filter bubbles).

## I. Introduction

Confirmation bias has been a topic of great interest among researchers across disciplines for several decades. Defined as "the tendency to acquire or process new information in a way that confirms one's preconceptions and avoids contradiction with prior belief" [1], it is considered one of many ways human judgment and decision-making deviate from neoclassical rationality. As such, it has sparked interest within various disciplines, such as behavioral economics, experimental psychology, opinion dynamics, and political science. One of the reasons for such widespread engagement is its critical societal implications, from misinterpretation of evidence about vaccines [2], to racist behavior [3], to climate-change denial [4]. As a consequence, there is a tendency to assume that confirmation bias is harmful to both individual and social decision-making.

Research from the past decade, however, suggests that a confirmation bias can be beneficial to individual decision-making. To understand such a bias, experimental psychologists use simple reinforcement learning algorithms. Reinforcement learning (RL) has proven itself as a minimal model to explain how human and non-human agents make decisions in stable or fluctuating environments [5]. According to this model, agents produce and maintain estimates that reflect the expected values of the options they can choose. As they interact with their environment, agents update their estimates, $Q_t$, with a prediction error, $\delta_t$, which quantifies the difference between an option's outcome at a given time ($r_t$) and the same option's predicted outcome ($Q_t$). During the update, the prediction error gets multiplied by a coefficient called "learning rate", $\alpha$, before being added to the previous estimate:

$$Q_{t+1} = Q_t + \alpha(r_t - Q_t)$$

The learning rate represents the weight an agent gives to new information. In recent studies involving cognitive biases, researchers have modelled biased decision-making by using asymmetric learning rates—i.e., agents attribute different weights to different types of prediction errors [6]. In experimental psychology, one of the most widely used paradigms to assess human reinforcement learning is the two-armed bandit task, since it is a very simple equivalent to many ecological choice scenarios. In this task, participants undergo a series of trials. At each trial, they make a choice between two options and get feedback—e.g., they get a reward in the form of food or money, or a punishment such as less money, or less food. On average, one of the options is more rewarding than the other, and the participants' goal is to learn the task's statistics so as to maximize rewards. Behavior on a two-armed bandit task can be accounted for using simple RL models. In the past decade, studies using this paradigm have shed light on a "positivity bias", whereby humans—as well as monkeys and mice—attribute more weight to positive than to negative prediction errors [7–10]. In humans, it appeared that this positivity bias was just a manifestation of a more complex confirmation bias: when they receive factual feedback—i.e., outcomes stemming from what an agent has chosen—, agents tend to give more weight to positive prediction errors; but, when it comes to counterfactual feedback—i.e., outcomes stemming from what an agent *could have chosen*, but has not—agents tend to dismiss positive prediction errors in favor of negative ones, which confirm that one was better off not choosing that option [11, 12]. Finally, Chambon *et al.* (2020) demonstrated that, when

human participants are forced to match a computer's decisions—i.e., they are forced to select the option the computer indicates to them —, they show no bias; a confirmation bias arises, however, when participants make their own decisions. Commenting on this finding, the authors suggest such a "choice-confirmation bias" could have emerged in order to reinforce the actions that are the most likely to satisfy an agent's needs—that is, free actions [13]. Besides this evidence, the potential adaptive value of positivity and confirmation biases has been stressed by several other authors. For instance, a few simulation studies suggest that, across a wide range of environments, virtual RL agents with a positivity bias [14, 15] or with a confirmation bias [16, 17] perform better than their counterparts. Therefore, confirmation bias seems to enhance individual decision-making rather than harm it.

Human and non-human animals, however, did not evolve in isolation. From hunting and gathering to purchasing products online, most of our decisions are made in a social context. To assess an object's value, we consider information we get from our peers. Nonetheless, it remains unclear whether a confirmation bias benefits decision-making in a social context. In the field of opinion dynamics, which is concerned with opinions rather than decisions, researchers have hinted at such a bias' effects at the collective level. In particular, confirmation has been linked to the emergence of problematic phenomena, such as echo chambers and polarization [18, 19]. A recent opinion-dynamics study, on the other hand, claims that a moderate amount of confirmation bias improves group learning, whereas a "strong" confirmation bias harms group consensus and leads to polarization [20]. In opinion dynamics, however, confirmation can be modelled in many different ways, and studies still lack a unified theoretical setup [18]. By contrast, RL—and its extension to the collective domain, multi-agent RL (MARL)—provides researchers with a cohesive and empirically grounded framework. Yet, glancing at the MARL literature reveals diverse algorithms that account for outcome-valence sensitivity in a collective setting. Among such algorithms, Lenient Learning [21], Hysteretic Q-learning [22], Win-or-Learn-Fast Policy Hill-Climbing [23], and variants of Frequency Maximization Q-learning [24, 25], have been shown to improve agents' convergence and cooperation in coordination games. Another recent RL study demonstrated that confirmation bias caused polarization in a group of agents playing the equivalent of a two-armed bandit task [26]. In that study, however, the agents were not connected—i.e., they were not directly observing their peers' actions and the resulting feedback. Yet, in our daily lives, we often base our decisions on the latter. For instance, if I want to get ice cream in one of two different shops standing next to each other, I may observe other customers' actions; if a customer enters the first shop and exits with mediocre-looking ice cream, I will be more likely to choose the second shop. Therefore, understanding how confirmation bias impacts decision-making when agents are connected—i.e., when they can observe their peers' actions and use this information to improve their own decisions—would constitute a significant step towards characterizing this bias in a more realistic setup. To the best of our knowledge, this issue has so far remained unaddressed in RL.

In this study, we hypothesize that the individual benefits of confirmation bias spill over to the collective level. We therefore expect that, in fully connected agent networks, confirmation bias has a positive impact on performance. As far as individual decision-making is concerned, Tarantola *et al.* suggest that such a bias improves performance as long as individuals do not dismiss counterfactual feedback [16]. In a collective setting, observing other agents' actions and rewards is one such way of keeping an eye on unchosen options. Indeed, our peers provide us with data about environmental states we do not observe ourselves, hence giving us a clearer picture of alternative choices. Therefore, we also hypothesize that confirmation bias' positive impact on performance will be amplified by the number of agents one is connected to.

To test our hypotheses, we build on previous results from experimental psychology and combine RL models with agent-based modelling. In particular, we assess the impact of a confirmation bias on RL agents' collective performance in a two-armed bandit task. The agents learn to solve the task in a social context: modelled as nodes in a fully-connected graph, they observe the other agents' actions and rewards. We seek to investigate how resource scarcity, group size, and bias strength modulate the confirmation bias' effect on performance.

In line with our hypotheses, we show that a confirmation bias benefits group learning across a wide range of resource-scarcity conditions and that this positive effect gets stronger with group size. In abundant environments, however, a confirmation bias harms performance for small groups of agents, and this effect is amplified by bias strength. Investigating the mechanisms whereby this effect occurs, we discover that, past a critical bias strength, resource abundance favors the emergence of two different performance regimes, one of which is suboptimal. Moreover, we show that this bifurcation towards a high- and a low-performance regime comes along with polarization in small groups of agents. This leads us to uncover the existence of an optimal bias strength for each condition.

## II. Results

### A. Confirmation bias enhances collective performance compared to other bias types

Our first aim was to assess whether confirmation bias provides groups of different sizes with an advantage over unbiased collectives, and over collectives in which agents have a disconfirmation bias—i.e., in which agents attribute a higher learning rate to disconfirmatory vs. confirmatory information. To this end, we studied how agents' performance on a two-armed bandit task—measured as average collected payoff per agent per trial—varies with bias type, group size, and resource scarcity. Thus, we simulated virtual agents performing a two-armed bandit task across various conditions, and observing every other agent's actions and rewards according to a model that we designed, the Collective Asymmetric Reinforcement Learning (CARL) model (see Section IV A).

In this model, agents discriminate between confirmatory and disconfirmatory prediction errors. An instance of confirmatory prediction error is when an agent, $i$, chooses option $A$ at trial $t$, and gets an outcome of $+1$. Since outcomes are either $+1$ or $-1$, and since an agent's Q-values are bounded between $-1$ and 1, getting an outcome of $+1$ (*resp.* $-1$) necessarily generates a positive (*resp.* negative) prediction error. Another instance of confirmatory prediction error is when, at $t$, $i$ sees another agent, $j$, get an outcome of $-1$ after choosing the alternative option, $B$. Conversely, a disconfirmatory prediction error can stem from agent $i$ obtaining an outcome of $-1$ from the option it has chosen; or from seeing agent $j$ obtain an outcome of $+1$ from the alternative option. In the CARL model, confirmatory and disconfirmatory prediction errors are not updated with the same learning rate. Typically, when agents have a confirmation bias, $\alpha^+$, the learning rate that updates confirmatory prediction errors, is higher than $\alpha^-$, the learning rate that updates disconfirmatory prediction errors.

Since information is always confirmatory or disconfirmatory with respect to a choice—i.e., independently from how much an agent values the chosen option —, the bias we study is technically a choice-confirmation bias. Throughout the paper, we will simply refer to it as "confirmation bias", which is also consistent with previous usage of the term in experimental literature and simulation studies on the topic [9, 16, 17].

In our first series of simulations, a confirmation bias corresponds to learning rates $\alpha^+ = 0.15$, and $\alpha^- = 0.05$; a disconfirmation bias to $\alpha^+ = 0.05$, and $\alpha^- = 0.15$; and no bias, to $\alpha^+ = \alpha^- = 0.1$. Additionally, we introduce "bias strength" as the ratio $b = \frac{\alpha^+}{\alpha^-}$. Since we sought to

investigate how bias type's effect was modulated by group size, we simulated groups of 1 to 20 agents. Our simulations were performed in three different scarcity conditions: a poor environment, in which both arms' reward probabilities, $p_1$ and $p_2$, are low ($p_1 = 0.3$, and $p_2 = 0.1$); a rich environment, in which they are high ($p_1 = 0.9$, and $p_2 = 0.7$); and a mixed environment, in which they are intermediate ($p_1 = 0.6$, and $p_2 = 0.4$). Finally, agents make choices using a softmax policy with inverse temperature $\beta$ (see Section IV A). $\beta$ quantifies the agents' tendency to explore vs. exploit both options; the lower $\beta$, the higher exploration. In all our simulations, we set all agents' inverse temperatures to $\beta = 4$, based on fitted parameters from previous experimental studies [9, 11, 13].

For each combination of conditions, we measured the collected payoff, which we averaged over agents, trials, and simulations. Then, we plotted it as a function of group size (Fig 1A–1C). In order to compare collective performance—i.e., performance that is averaged over all agents in the group—with individual performance—i.e., performance of one single agent within the group—, we also plotted mean collected payoff per trial for one arbitrarily-chosen agent (see S1A–S1C Fig).

From Fig 1A–1C, one observes first that, across most scarcity and group-size conditions, confirmation-biased agents' performance is significantly higher than unbiased agents' and disconfirmation-biased agents' performance ($p < 0.01$, see confidence intervals). The exceptions are $n < 4$ agents in rich environments and solitary agents ($n = 1$) in mixed environments. This means that, in most conditions, a confirmation bias allows agents to collect more rewards on average, thus conferring an advantage over other bias types. In addition, disconfirmation-biased agents perform almost always worse than unbiased agents, suggesting that a disconfirmation bias harms decision-making in general.

Second, in all resource-scarcity conditions, confirmation-biased agents' performance increases with group size. In a rich environment, in particular, increasing group size cancels the advantage small groups—i.e., 1 or 2 agents—of unbiased agents have over small groups of confirmation-biased agents. This suggests that confirmation-biased agents benefit from receiving information from a larger number of neighbors. In comparison, unbiased agents'

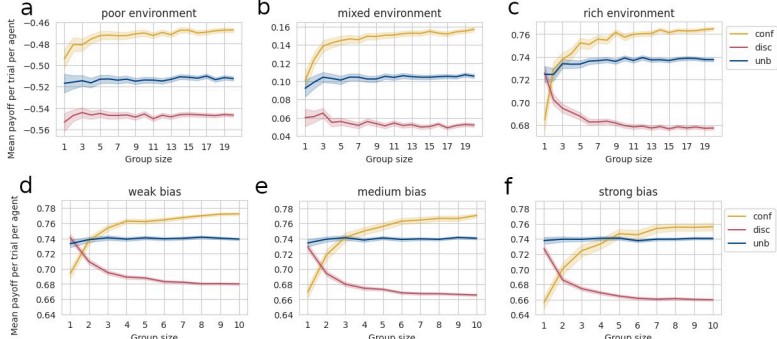

**Fig 1. Mean collected payoff per trial per agent.** A-C: as a function of group size, bias type, and resource scarcity. A: poor environment: $p_1 = 0.3$, $p_2 = 0.1$. B: mixed environment: $p_1 = 0.6$, $p_2 = 0.4$. C: rich environment: $p_1 = 0.9$, $p_2 = 0.7$. Yellow curve: confirmatory agent, $b = 3$; blue curve: unbiased agent; red curve: disconfirmatory agent, $b = \frac{1}{3}$. Transparent zone denotes 95%-confidence intervals. For each combination of conditions, $N_{simu} = 500$ simulations were run. Within each simulation, agents perform the task over $N_{trials} = 100$ trials—i.e., they activate the bandit 100 times. D-F: as a function of group size, bias type, and bias strength in a rich environment. D: weak bias: $b = 3$ for confirmatory agents, $b = \frac{1}{3}$ for disconfirmatory agents. E: medium bias: $b = \frac{17}{3}$ for confirmatory agents, $b = \frac{3}{17}$ for disconfirmatory agents. F: strong bias: $b = 9$ for confirmatory agents, $b = \frac{1}{9}$ for disconfirmatory agents. For each combination of conditions, $N_{simu} = 500$ simulations were run. Within each simulation, agents perform the task during $N_{trials} = 200$ trials, which allows observing the effect of bias on both the learning phase and the stationary state after convergence.

performance seems to be always unaffected by group size. As for disconfirmation-biased agents, increasing group size is detrimental to performance in rich environments but does not seem to have a significant effect in other resource-scarcity conditions.

Plots of individual performance (see S1A–S1C Fig) exhibit the same patterns, indicating that bias type, resource scarcity, and group size have a similar impact on every single agent in the group.

## B. Stronger confirmation bias impedes decision-making in a rich environment

Our previous analysis suggests that a confirmation bias, although beneficial in general, could harm small groups' performance in a rich environment. Investigating this effect, we wondered how small groups would perform if the bias got stronger. Therefore, we varied the bias strength $b$—defined as the learning-rates ratio $\frac{\alpha^+}{\alpha^-}$—and studied its impact, in a rich environment, on the comparative performance of confirmation-biased vs. unbiased agents. In particular, we looked at the intersection point between the unbiased curve and the confirmatory curve, since this intersection point indicates the group size at which confirmatory agents start to perform better than unbiased agents. Furthermore, we kept comparing with disconfirmation-biased groups.

To this end, we simulated virtual agents performing a two-armed bandit task in a rich environment, and explored three bias-strength conditions: weak, medium, and strong. In the weak condition, $b = \frac{\alpha^+}{\alpha^-} = \frac{0.15}{0.05} = 3$ for confirmation-biased agents, while $b = \frac{\alpha^+}{\alpha^-} = \frac{0.05}{0.15} = \frac{1}{3}$ for disconfirmation-biased agents. In the medium condition, $b = \frac{\alpha^+}{\alpha^-} = \frac{0.17}{0.03}$ for confirmation-biased agents, while $b = \frac{\alpha^+}{\alpha^-} = \frac{0.03}{0.17}$ for disconfirmation-biased agents. In the strong condition, finally, $b = \frac{\alpha^+}{\alpha^-} = \frac{0.18}{0.02} = 9$ for confirmation-biased agents, while $b = \frac{\alpha^+}{\alpha^-} = \frac{0.02}{0.18} = \frac{1}{9}$ for disconfirmation-biased agents. In all bias-strength conditions, we compare confirmation- and disconfirmation-biased agents with unbiased agents, whose bias strength is $b = \frac{\alpha^+}{\alpha^-} = \frac{0.1}{0.1} = 1$. Finally, we also vary group size: for each combination of conditions, we simulated groups of 1 to 10 agents.

In Fig 1D–1F, we plotted collected payoff—averaged over agents, trials, and simulations—as a function of group size. In order to compare collective performance with individual performance, we also plotted the average collected payoff per trial for one arbitrarily-chosen agent, as a function of group size (see S1D–S1F Fig).

From Fig 1D–1F, it appears that, in all bias-strength conditions, increasing group size does not have the same effect on confirmatory vs. unbiased vs. disconfirmatory agents. Indeed, while confirmatory agents' performance increases with group size, disconfirmatory agents' performance decreases. Unbiased agents' performance, in comparison, does not seem to be affected by an increase in group size. This suggests that, across all bias-strength conditions, confirmatory agents benefit from being in a bigger group, whereas such an increase in group size is detrimental to disconfirmatory agents. In addition, disconfirmatory agents perform almost always worse than unbiased agents, suggesting that a disconfirmation bias harms decision-making whatever its strength.

Finally, we compared the intersection points between the blue curve and the yellow curve for different bias strengths. In all bias-strength conditions, indeed, smaller groups of confirmatory agents perform worse than their unbiased peers. When group size increases, confirmatory agents increase their payoffs, until they outperform unbiased agents. However, the group size at which they take the lead depends on bias strength: with a weak bias, the intersection point between the blue and yellow curves lies at group size 2 (Fig 1D); with a medium bias, at size 3 (Fig 1); and, with a strong bias, between sizes 4 and 6 (Fig 1F). This means that, the higher the

bias strength, the higher the required group size for confirmation-biased agents to outperform unbiased agents. Therefore, although increasing bias strength has a detrimental impact on confirmatory agents, increasing group size seems to counterbalance this negative effect on performance. This shows that, in a rich environment, a strong confirmation bias impedes decision-making without harming it completely: its negative impact is not robust against an increase in group size. All in all, this suggests that larger groups can tolerate stronger biases.

Plots of performance for one agent within the group (see S1D–S1F Fig) exhibit the same patterns, indicating that bias strength, resource scarcity and group size have a similar impact on every single agent in the group.

## C. Varying bias strength gives rise to bimodality in Q-value-gap distributions

In the previous section, we showed that confirmation bias impedes performance in a rich environment. As a consequence, we sought to understand how such an effect arises. A previous simulation study showed that confirmation bias enhances individual performance by increasing the gap between the Q-values an agent attributes to both options [17]. Glancing at the softmax formula, one easily understands that, as long as the best option's Q-value is kept higher than the worst option's, increasing their difference leads to a higher probability of choosing the best option, hence counteracting the effect of noise. Thus, Q-value gap is directly related to performance. In this section, we study the effect of bias strength on Q-value gap distributions, in order to determine whether the confirmation bias' impeding effect can be explained by a decrease in Q-value gap.

To this end, we simulated virtual agents performing a two-armed bandit task in a rich environment, and explored three bias-type conditions: unbiased, weak confirmation bias, and strong confirmation bias. In the unbiased condition, agents have a bias strength $b = \frac{\alpha^+}{\alpha^-} = \frac{0.1}{0.1} = 1$; in the weak confirmation bias condition, $b = \frac{\alpha^+}{\alpha^-} = \frac{0.15}{0.05} = 3$; in the strong confirmation bias condition, $b = \frac{\alpha^+}{\alpha^-} = \frac{0.18}{0.02} = 9$. We aim to understand why larger confirmatory groups outperform smaller confirmatory groups in a rich environment. Therefore, we ran simulations with two group sizes that seemed sufficiently far apart for the difference to qualitatively affect the final outcomes—i.e., $n = 2$ and $n = 5$.

In Fig 2A–2C, we plotted one agent's final-Q-value-gap ($\Delta Q_f$) distributions over 1000 simulations, and compared these distributions across group sizes and bias types. A first striking observation is that bias strength seems to have an impact on the distributions' shape. Indeed, when $n = 2$ (blue), Q-value-gap distribution goes from unimodal (unbiased agents, Fig 2A), to weakly bimodal (weakly confirmatory agents, Fig 2B), to strongly bimodal (Fig 2C). This means that, when agents are unbiased, performance is rather homogeneous across simulations. When they are equipped with a confirmation bias, however, two different regimes of performance emerge: in some simulations (leftmost blue peak), weakly biased agents perform rather poorly, whereas in others (rightmost blue peak), they perform quite well. As for strongly biased agents, the separation between the two performance regimes is quite extreme: in some simulations (leftmost blue peak), agents perform very poorly—negative Q-value gaps indicate that, in a non-negligible fraction of the simulations, they even tend to choose the worst option more often than the best—whereas in others (rightmost blue peak), they perform very well.

But Fig 2A–2C also suggests that, when $n = 5$ (pink distributions), bimodality tends to disappear. Thus, Fig 2B shows a single peak—i.e., a single performance regime—for weakly biased agents. For strongly biased agents (Fig 2C), on the other hand, bimodality remains, but most of the distribution's weight lies on higher values of $\Delta Q_f$, suggesting that simulations resulting in low performance have only a marginal effect on mean collected payoffs. This means that,

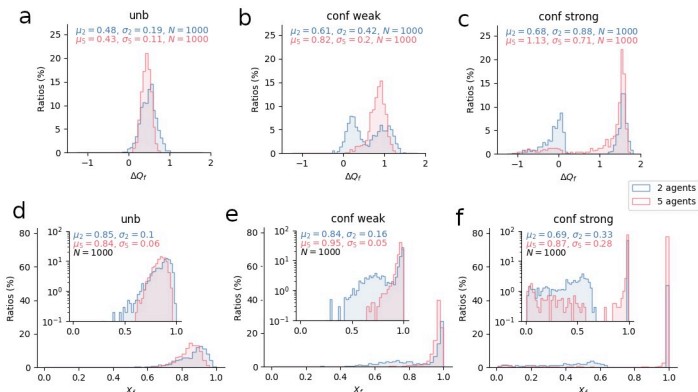

**Fig 2. Final-Q-value-gap and final-behavior-profile distributions for one agent in a group as a function of group size, bias type, and bias strength, at the end of a sequence of 1000 trials.** A-C: Q-value-gap distributions for one agent in a group with no bias (A), with a weak confirmation bias $b = 3$ (B), with a strong confirmation bias $b = 9$ (C). D-F: Behavior-profile distributions for one agent with no bias (D), with a weak confirmation bias $b = 3$ (E), with a strong confirmation bias $b = 9$ (F); inserted plots show the same distributions, but with the $y$-axis in log-scale. Blue distributions: $n = 5$; pink distributions: $n = 2$. For each plot, number of bins was set to 50. For each combination of conditions, $N_{simu} = 1000$ simulations were run. Within each simulation, agents perform the task during $N_{trials} = 1000$ trials. This high number of trials has been chosen to get a precise estimate of Q-value gaps after convergence. Q-value gap was measured at each simulation's very last trial, and for one arbitrarily-chosen agent in the group (i.e., "agent 0" in the simulations). Plotted distributions were obtained for that agent over the 1000 simulations. Behavior-profile distributions were obtained by applying the softmax function to previous Q-value data.

with increasing group size, confirmatory agents are less likely to end up with a small or negative Q-value gap, thus exhibiting high performance most of the time. These results echo the impeding effect found in Section II B.

Nonetheless, studying bias strength's impact on $\Delta Q_f$-distributions does not, on its own, explain the bias' impeding effect. To convince oneself, one only needs to look at the blue distributions' means: when a group of two is equipped with a strong confirmation bias (Fig 2C), the distribution's mean is highest despite a non-negligible fraction of simulations that end with very poor performance. This is not in line with our results from the previous section, which demonstrated that, on average, groups of two unbiased agents collect more rewards than their strongly biased counterparts. An explanation can be found in the softmax function, which relates Q-value gap $\Delta Q$ to policy or "behavior profile" $X$ [27]. Indeed, this function is bounded and non-linear. Therefore, if the mean of a given $\Delta Q_f$-distribution 1 is higher than the mean of another $\Delta Q_f$-distribution 2, it does not entail that the mean of the corresponding $X_f$-distribution 1 is higher than the mean of the other corresponding $X_f$-distribution 2. When one applies the softmax function to each data point in our $\Delta Q_f$ distributions, one obtains $X_f$ distributions (Fig 2D–2F)—i.e., distributions of an agent's final behavior profiles over 1000 simulations. Although bimodality is less visible on these graphs, it still exists, and the low-performance regime's negative contribution to mean performance becomes very clear in the two-agent case: mean behavior profile equals 0.69 (Fig 2F), which is noticeably lower than 0.85, the mean behavior profile for unbiased agents (Fig 2D). Therefore, bimodality in $\Delta Q_f$ distributions translates into lower mean probabilities of choosing the best option, which, in turn, translates into lower mean performance.

Finally, Fig 2D–2F shows that, in a rich environment, a weak confirmation bias seems very beneficial to groups of five agents. Indeed, mean behavior profile is 0.95 (Fig 2E), significantly higher than unbiased (Fig 2D) and strongly biased (Fig 2F) agents' mean behavior profiles (0.84 and 0.87, respectively). This suggests the existence of an optimal, group-size dependent, bias strength.

## D. Varying bias strength gives rise to different regimes of performance and polarization

In the previous section, we showed that, in a rich environment, confirmation bias gives rise to bimodality in the agents' final-Q-value gap distributions, and that increasing group size attenuates this bimodality. Nonetheless, using only three bias-strength conditions gives a partial picture of the performance regimes, and one is left to wonder at which bias strength, exactly, the two regimes emerge. To address this, we visualize Q-value gap distributions as a function of bias strength, where bias strength slides from $b = 1$ to $b = 5$. Thus, $b = 1$ corresponds to no bias, whereas $b > 1$ corresponds to different confirmation-bias strengths. In all bias-strength conditions, $\alpha^- = 0.1$, and $\alpha^+ = b \cdot \alpha^-$. Moreover, we studied the impact of group size and resource scarcity on the distributions. To this end, we used the first experiment's poor, mixed, and rich environments, as well as the previous experiment's group sizes—i.e., groups of 2 and 5 agents.

Since bimodality can be easily detected by looking at the smoothed distributions' maxima, we decided to plot these maxima across bias strengths. For this purpose, we first computed the distributions' kernel density estimators (KDEs), and located their plots' maxima automatically. In Fig 3A–3C, the number of points corresponding to a given bias strength is the number of maxima that the $\Delta Q_f$ distribution exhibits for this bias strength.

By comparing Fig 3's three different plots, one immediately observes that, contrary to the mixed and rich conditions, the poor condition does not cause bimodality to emerge in the distributions (Fig 3A). Rather, for both groups of 2 and 5 agents, $\Delta Q_f$ distributions exhibit one maximum, which value increases with bias strength. Moreover, this maximum is always positive and higher than the difference between both options' expected values—i.e., 0.4. This means that a poor environment is always conducive to a single, high-performance regime, whatever the bias strength and the group size. Moreover, this suggests that, in a poor environment, performance increases with bias strength, although, to be sure, one would need to convert the $\Delta Q_f$ distributions to $X_f$ distributions and compute their means.

Turning to Fig 3A and 3C, one notices the emergence of bimodality past a certain bias strength. As can be seen by comparing 2 and 5-agent plots, bimodality is more pronounced for groups of 2. Interestingly, resource scarcity seems to have an impact on the maxima's locations: while in a mixed environment, negative maximum locations arise past $b = 3$, a rich environment gives rise to maximum locations that approach 0 but do not become negative. This means that, although both resource-scarcity conditions lead, past a certain bias strength, to the emergence of low-performance regimes, the latter are quite different. In the mixed condition, indeed, negative values indicate that agents favor the worst option, hence making "wrong" choices most of the time; in the rich condition, on the other hand, values close to zero suggest agents tend to favor the best and the worst option almost equally, hence following a nearly random policy. The 2-agent plots, Fig 3B and 3C, confirm our previous observation that increasing group size tends to attenuate bimodality, as points located in the low-performance regime are much more transparent than for groups of 2 agents—which signals lower maximum values. This is most striking in the rich condition (Fig 3C), which, for 5 agents, gives rise to a strong bias towards the high-performance regime. For 2 agents, in comparison, the low-performance regime's maxima are higher, indicating that a greater fraction of simulations end with poor performance. This result echoes our previous findings, which demonstrate that, in a rich environment, adding agents to the group makes a noticeably positive difference to collective performance. Finally, we plotted the full final-Q-value-gap distributions for both group sizes as heatmaps (see S2 Fig). These show the same patterns and bifurcations, confirming the emergence of distinct high- and low-performance regimes.

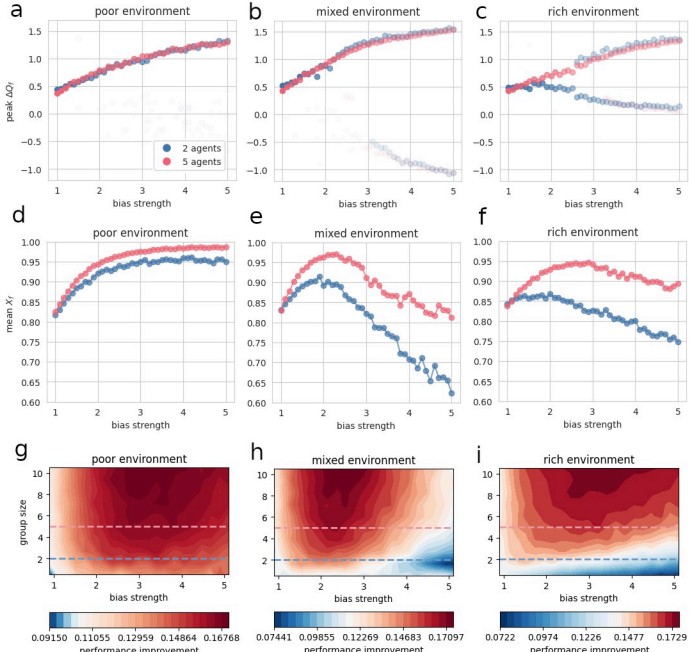

**Fig 3. Performance regimes and optimal bias strength.** A-C: Local maxima of Q-value-gap distributions' KDE plots for one agent in a group, as a function of bias strength, group size, and resource scarcity, at the end of a sequence of 1000 trials. Blue plot: 2 agents; Pink plot: 5 agents. Transparency denotes peak height divided by the highest peak's height in the plot. For each combination of conditions, $N_{simu}$ = 1000 simulations were run. In each simulation, agents perform the task during $N_{trials}$ = 1000 trials, so as to obtain precise estimates of Q-value gaps after convergence. Q-value gap was measured at the very last trial, focusing on one agent within the group (i.e., the agent that was labelled "agent 0" in the simulations). For each combination of conditions, a kernel density estimator (KDE) plot of the Q-value-gap distribution was computed. From these KDE plots, local maxima's coordinates were automatically extracted. Finally, for each group size and resource-scarcity condition, the peaks' $x$-coordinates (which correspond to the Q-value gaps for which the distribution is at a local maximum) were plotted across bias strengths. To distinguish between high and low peaks, transparency was increased with decreasing peak height. D-F: Mean behavior profile for one agent in a group, as a function of bias strength, group size, and resource scarcity, at the end of a sequence of 1000 trials. Blue plot: 2 agents; Pink plot: 5 agents. To obtain these plots, we converted each data point from the previous Q-value-gap distributions to a behavior profile *via* the softmax formula. We then computed the mean of every subsequent behavior-profile distribution. Mean behavior profile is strongly correlated with performance and performance improvement (see Fig 3G-I). G-I: Performance improvement per agent per trial, as a function of bias strength, group size, and resource scarcity, during the last 100 trials of a sequence of 200 trials. Dashed lines indicate how to relate G-I with D-F: performance improvement is strongly correlated with mean behavior profile for 2 agents (blue) and 5 agents (pink). For each combination of conditions, $N_{simu}$ = 500 simulations were run. In each simulation, agents perform the task during $N_{trials}$ = 200 trials. Performance improvement denotes the difference between actual performance and expected performance when agents follow a random policy (see Section IV B).

From $\Delta Q_f$ distributions, we can compute mean Q-value gap across bias strengths, and look at its local optima. Similarly, we can convert $\Delta Q_f$ distributions to $X$ distributions, compute and analyze the mean behavior profile versus bias strength. S3 Fig shows mean Q-value gap as a function of bias strength in each resource-scarcity condition, for 2 and 5 agents. The corresponding mean behavior profiles are shown in Fig 3D–3F. Interestingly, the mean behavior profiles demonstrate the existence of an optimal bias, which maximizes performance for both group sizes. Although slightly different, these optimal values are located at a moderate level of bias strength, between $b = 2$ and $b = 3$. Past these optima, in mixed and rich environments, performance keeps decreasing. In a poor environment, on the other hand, performance increases with bias strength for both group sizes, although it seems to reach a plateau eventually (Fig 3D–3F). This is correlated with the absence of a low-performance regime in the poor environment.

Fig 3G–3I shows mean performance improvement—defined as the difference between actual performance and expected performance for a random policy (see Section IV B)—as a function of bias strength and group size for groups of 1 to 10 agents. As such, it summarizes our results well: performance improvement is highest for intermediate levels of bias strength; in mixed and rich environments, performance decreases dramatically for small groups past a certain bias strength (whereas, in poor environments, performance improvement is more robust across bias strengths and group sizes); in general, larger groups tolerate stronger biases.

We demonstrated that, under specific circumstances, CARL's learning dynamics cause individual performance to split into two different regimes. However, the question remains whether a split occurs within the collective as well: during the same run, do agents converge towards different regimes, or towards the same one? In other terms, do agents exhibit polarization or consensus? To answer this question, we visualized the distribution of differences between agents' final Q-value gaps. For every group-size and resource-scarcity conditions, we plotted the distributions of differences between a focal agent's and the other agents' final Q-value gaps across 1000 simulations, as a function of bias strength. Interestingly, we found that, in a group of 2 agents, mixed and rich environments always give rise, past a certain bias strength, to polarization in the final Q-value gaps (Fig 4A). Moreover, the higher the bias strength, the more polarized the agents. In poor environments, on the other hand, agents remain in a consensual state for all bias strengths (Fig 4A). Fig 4B suggests that adding agents to the group tends to reestablish consensus: in mixed and rich environments, most simulations end with the first and the second agents converging towards the same Q-value gaps, although a few simulations do end, past a certain bias strength, in a polarized state. Comparing any agent with the focal one gives rise to a similar pattern (see S4 Fig). Such polarization may explain

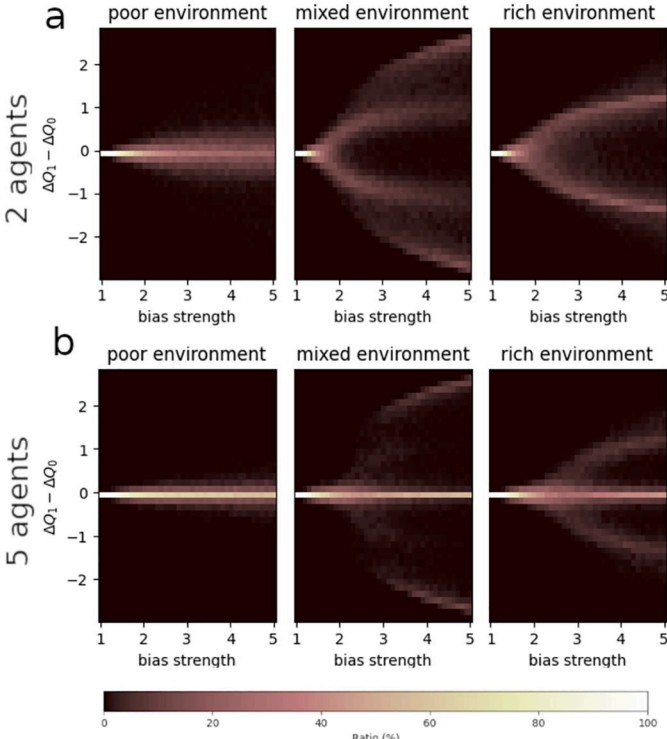

**Fig 4. Distributions of differences between agent 0's and agent 1's Q-value gaps as a function of bias strength.** A: In a group of 2 agents; B: In a group of 5 agents.

small groups' poor performance in abundant environments, and the decrease in performance as bias strength increases. It also suggests that, by reestablishing consensus, larger groups improve their performance on the task.

To understand why polarization emerged in groups of two agents, we devised a deterministic version of our CARL model (see Section IV A4), and simulated this model over 500 trials in poor, mixed, and rich environments, with various initial conditions (see S5 Fig). This model allowed us to visualize deterministic trajectories of both agents' Q-value gaps. Interestingly, S5 Fig shows that, in mixed and rich environments, consensus is a fixed point of the system (S5C and S5E Fig). However, it is likely an unstable one: when agents do not start with strictly identical Q-values, their final Q-value gaps diverge (S5D and S5F Fig). In a poor environment, by contrast, slightly different initial conditions do not lead final Q-values to dramatically diverge. In our agent-based simulations, thus, groups of two agents never converge, in mixed and rich environments, towards consensus, because of noise: although agents start with strictly identical Q-values, stochasticity induced by the task or by their policies makes them deviate from the unstable fixed point, so that they inevitably converge towards the stable, polarized state. Although we did not develop a deterministic model for groups of five agents—as adding agents would complexify the equations —, our agent-based simulations with 5 agents suggest that increasing group size eventually reestablishes consensus as a stable state.

Finally, additional simulations with higher temperature—i.e., lower values of $\beta$—were carried out to study whether polarization would emerge when agents are more exploratory (see S1 Text). We found out that, under a critical value of $\beta$—which differs depending on resource scarcity—polarization disappears. This is because, when agents are more exploratory, they get more opportunities to gather confirmatory information from alternative options. This prevents them from getting trapped in incorrect attractors. Interestingly, these additional results show that, although polarization disappears for lower values of $\beta$, small groups of confirmatory agents still exhibit poorer performance than their unbiased counterparts in a rich environment. Further analysis (see S1 Text) reveals that small groups behave like optimistic agents: they inflate both options' Q-values. A previous study [14] demonstrated that such optimism is beneficial in poor environments but not in abundant ones, as Q-values' inflation saturates when approaching 1. For this reason, small groups of confirmatory agents end up with smaller Q-value gaps in rich environments, which negatively impacts their performance. In larger groups, confirmatory agents still inflate the best option's Q-values, but the sampling of the alternative option by another agent is more likely. Therefore, agents get more opportunities to deflate the worst option's Q-value, which results in a larger Q-value gap, and thus in higher performance. A detailed explanation of these effects is provided in S1 Text.

## III. Discussion

In the past years, researchers have witnessed a surge of interest in the confirmation bias at the level of individual decision-making. Evidence has shown that humans exhibit such a bias, and that it could improve performance in a wide range of environments. Nevertheless, confirmation bias, as framed in a reinforcement learning framework, has not yet been thoroughly investigated at the collective level. Yet, most of our decisions are based on information we collect from our peers, which makes it crucial to understand how cognitive biases influence decisions made in a social context. Our study constitutes a step in that direction by exploring the impact of such a bias on decision-making under social influence in a simple choice task. To this end, we used agent-based simulations, and varied bias type, bias strength, group size, and resource scarcity. We first asked what would be the impact of a confirmation bias on performance, compared to no bias or a "disconfirmation" bias. Then, we sought to understand bias strength's

effect on performance, and how this effect is modulated by group size and resource scarcity. This led us to investigate how bias strength impacts final Q-value gaps, and how it can lead, under certain conditions, to polarization within the group.

Overall, we find that groups with a confirmation bias exhibit higher performance than unbiased groups, and than groups with a disconfirmation bias. Such higher performance is found across most resource conditions and group sizes. In abundant environments, however, a confirmation bias can be detrimental to smaller groups. The stronger the bias, the higher the required group size to compensate for this disadvantage. In other terms, larger groups can tolerate stronger confirmation biases. Finally, we find that a confirmation bias gives rise to a bifurcation between regimes of high and low performance, which may explain lower collected payoffs in abundant environments. This points towards an optimal level of confirmation bias for decision-making in a social context. In mixed and rich environments, we discovered that individual agents can end up in one of two different performance regimes past a certain bias strength. In groups of 2 agents, this bifurcation comes with a polarized state, whereby each agent performs in a different regime. This means that, in these conditions, overall mediocre performance is underlain by polarization between a high-performing agent and a poorly-performing agent.

In a nutshell, our results suggest that a confirmation bias improves learning of action-outcome contingencies, and that this improvement is robust across resource scarcities. Performance robustness in larger groups of agents, in particular, hints at the confirmation bias' possible adaptive value in a social context.

Despite poor performance for small groups in mixed and rich environments, groups with a confirmation bias exhibit overall higher performance than unbiased and disconfirmatory groups. This suggests that a confirmation bias can be advantageous to decision-makers in a social context, and that this advantage is robust across various resource scarcities and group sizes. These results are in line with our initial prediction that a confirmation bias can boost collective performance on a two-armed bandit task in a social context. Such an improvement in performance, across various scarcity conditions, has been demonstrated at the individual level [17]. Our results extend these previous findings to the collective level.

As previously mentioned, there are exceptions to the confirmation bias's advantage over disconfirmation or no bias: in abundant environments, confirmation can be detrimental to smaller groups. The stronger the bias, the higher the required group size to compensate for this disadvantage. These results suggest that, in a social context, agents need to accumulate more information in order to compensate for a strong confirmation bias' negative effect. Accumulating more information is done by gathering and averaging data from more individuals—i.e., increasing group size. This also points to the idea that a confirmation bias is not advantageous *per se*: if it is too strong, it is actually detrimental to performance. Therefore, at the collective level, there are some contexts in which a confirmation bias can impede efficient decision-making. In our model, the single-agent case reduces to a simple positivity bias: the agent updates positive prediction errors with a higher learning rate than it does negative prediction errors. In short, there are no other agents to feed it counterfactual information. Cazé & van der Meer had previously shown that a positivity bias leads to poorer performance for a single agent in a rich environment [14]. But it was not obvious that a confirmation bias would also lead to poorer performance in groups of two or three agents. Similar results, however, have been obtained by Gabriel & O'Connor with an opinion-dynamics model, and independently from resource scarcity [20]. The authors implemented a "strong confirmation bias", and showed that, for small group sizes, strongest biases harm correct consensus. For larger group sizes, however, bias strength does not seem to make a difference.

Finally, we found that a confirmation bias gives rise to specific learning dynamics which, in particular, may explain lower performance in abundant environments, and which point towards an optimal level of confirmation bias for decision-making in a social context. Past a certain bias strength, bifurcations in final Q-value gaps emerge in mixed and rich environments, hinting at two different performance regimes. Further investigation revealed that this bifurcation came with the emergence of polarization within groups of 2 agents. Such polarization between high- and low-performing individuals within the group constitutes a mechanism whereby collective performance declines. However, supplementary analyses revealed that polarization was not the only causal factor for such performance drop. Indeed, small groups of confirmatory agents tend to behave like optimistic learners, yet optimism is only beneficial in environments where positive outcomes are rare [14]. When more agents are added to the group, the odds that a peer chooses the alternative option are higher; thus, agents get more opportunities to deflate the worst option's Q-value, and they end up with larger Q-value gaps.

It has been suggested that reduced sampling of previously punishing options, called "hot stove effect" [28], could be corrected by interdependent sampling processes [29] or conformist social learning [30]. Although this is reminiscent of the aforementioned results, there are two main differences between the hot stove effect and the drop in performance we underline. Indeed, the effects we observe in small groups of confirmatory agents are not due to reduced sampling *per se*, but to reduced *counterfactual* sampling. This reduced counterfactual sampling itself does not stem from previous punishments but, on the contrary, from a lack of punishment combined with a lack of sampling neighbors. Nonetheless, just like the cited studies about the hot stove effect, our model also makes use of a minimalist form of social influence to counteract sampling-induced errors. Together, these works allow further understanding of various kinds of sampling-induced errors, and of the way social learning helps correct them.

In the individual case, Lefebvre *et al.* already showed that a confirmation bias can lead learning dynamics towards an incorrect attractor [17]. They give an expression for a "critical bias", above which an agent's trajectory can end up in a correct or an incorrect attractor depending on its history. Our study suggests there is a similar mechanism at play with CARL, at least in fully-connected networks. In the social case, similar results have been obtained by Gabriel & O'Connor with an opinion-dynamics model [20], and independently from resource scarcity. The authors argue that a moderate confirmation bias actually serves group learning, and could have developed in the social environment in which humans evolved. In particular, they cite Mercier & Sperber (2017), who hold a similar view: according to them, confirmation bias leads disagreeing members of a group to stick to their respective positions, hence encouraging them to develop good arguments in favor of their preferred views, and thereby making final consensus better informed [31]. Our results suggest that a confirmation bias indeed benefits decision-making, but can be harmful above certain values. This echoes Lefebvre *et al.*'s conclusions at the individual level [17]. Designing an experimental version of our task and determining whether participants exhibit a moderate confirmation bias would be an interesting endeavor for future research.

## A. Limitations of the present study

One limitation of our study is that we consider a restricted set of environments; many other settings are left to be investigated. For instance, the question remains whether a confirmation bias improves collective performance on any multi-armed bandit task, as for example, two-armed bandit tasks with any contingencies or bandits with more than two arms. Agents in our study are identical and confronted with the same individual task. Situations where heterogeneous agents interact with each other, and actions directly impact the outcome for others, such

as in social dilemmas, present exciting and relevant cases for future research. For example, extending previous work investigating the conditions under which RL agents learn to cooperate in social dilemmas [32] is a concrete next step. Moreover, the environments we used in this study are all static; yet, in the everyday world, we make decisions in fluctuating circumstances. Since confirmation bias tends to make agents more conservative—by increasing Q-value gap and, hence, probability of choosing one's favored option—, one may expect that such a bias puts collectives at a disadvantage in fluctuating environments. This question could be an engaging avenue for future research.

Another limitation pertains to the way our agents gather information. First, it is assumed that each agent can observe every other agent's actions and rewards; this is unrealistic in large groups, as visual attention has been shown to have limited capacity [33]. The framework of partial observability could be used to model explicit environmental and perceptual uncertainty [34]. Second, our agents choose options and update estimates in a synchronic fashion; this is a simplification, as humans tend to accumulate evidence over time [35], and groups exhibit complex decision-making dynamics whereby individual decision timing depends on confidence [36], and selfishness [37]. Therefore, determining whether our findings hold for more realistic network structures—i.e., network structures that take into account the dynamics of attention—would be an exciting question to answer next. Third, Chambon *et al.* suggested that humans exhibit a confirmation bias when they are free to choose, but not when a computer makes choices for them [13]. In our setup, agents do decide freely, and their peers' choices provide them with counterfactual feedback. In nature, however, animals often gather information through an observational-learning phase before making a decision. It is unclear whether a confirmation bias would apply during purely observational learning—i.e., when the focal agent refrains from making a choice while it learns from the others' actions. This question should be answered experimentally. However, we believe our model could be very useful in formulating predictions regarding this question.

Finally, one should keep in mind that, in observed human behavior, confirmation bias could interact, or be confused, with other types of biases. For instance, Sugawara and Katahira showed that previous experimental results involving asymmetric learning could be explained, in fact, by a perseverance bias, or by a mix of perseverance and asymmetric learning [38]. In our study, we do not determine whether the performance curves are a characteristic signature of confirmation bias, or could be obtained with, e.g., perseverance. Designing a model that could help discriminate between perseverance bias and asymmetric learning, such as Sugawara and Katahira's Hybrid model but at the collective level, would be useful to future experimental work.

## B. Conclusion

By shedding light on the mechanisms underlying learning dynamics in collectives where agents exhibit a confirmation bias, our study suggests such a bias could have been adaptive in the social context in which humans evolved. Therefore, rather than being a sign of irrationality, human agents' tendency to seek confirmatory information may exist for a good reason. In an era that many have qualified as "post-truth", this realization could have important consequences. In particular, targeting problematic social phenomena, such as misinformation and polarization, may require that we focus less on confirmation bias *per se*, but rather emphasize other features, such as bias strength, network structure or inter-individual influence. Such a task calls for *complex methodological individualism* [39], which considers how individuals change and are changed by groups through social interactions. As technology has multiplied

the number and range of such interactions, placing our scientific endeavors at the interface between the individual and society may prove crucial to tackling contemporary challenges.

## IV. Methods

### A. Collective Asymmetric Reinforcement Learning model

Here, we introduce the Collective Asymmetric Reinforcement Learning (CARL) model, which describes reinforcement learning dynamics in a collective of virtual agents performing a two-armed bandit task. In this model, agents are equipped with two different learning rates, and it is assumed that both bandits return a payoff of either +1 ("reward") or −1 ("penalty"). In particular, the first bandit returns a reward with probability $p_1$, and the second bandit, with probability $p_2$. Therefore, the expected payoff for the different bandits $j$ is $2p_j − 1$, with $j = 1, 2$.

Within the collective, each agent tries to maximize its payoff using a Q-learning algorithm without discounting and different types of information: information resulting from its own actions, and information stemming from the other agents' actions. In addition, the algorithm distinguishes between confirmatory and disconfirmatory information, which we define below.

**1. Information types.** CARL assumes that $n$ agents ($n \in \mathbb{N}^*$) perform the same two-armed bandit task synchronously, during a number $T$ of time steps, or "trials" ($T \in \mathbb{N}^*$). At the end of each trial, each agent can observe not only its own action and payoff, but also every other agent's actions and payoffs. Thus, agents can be represented as nodes in a fully-connected graph, whose edges, $e_{ij}$, denote observation of agent $j$'s actions and payoffs by agent $i$. At the end of every trial $t$, $t \in [1, T]$, each agent updates its Q-values according to a modified Rescorla-Wagner rule, which takes other agents' actions and payoffs into account. Different learning rates are used by every agent to update confirmatory *vs.* disconfirmatory information, as defined in the following paragraph.

Let us denote $a_t^{i,c}$ agent $i$'s chosen bandit at trial $t \in [1, T]$ ($i \in [1, n]$). Let us call $\mathcal{R}_t^{i,c}$ the set of all agents who chose the same bandit as agent $i$ at trial $t$, and who obtained a reward (+1) from that bandit at trial $t$. Similarly, $\mathcal{P}_t^{i,c}$ denotes the set of all agents who chose the same bandit as agent $i$ at trial $t$, and who obtained a penalty (−1) from that bandit at trial $t$. If agent $i$ itself obtained a reward at trial $t$, then $i \in \mathcal{R}_t^{i,c}$. Otherwise, $i \in \mathcal{P}_t^{i,c}$.

Let $a_t^{i,u}$ be the bandit which agent $i$ did not choose at trial $t$. One may define two other sets, $\mathcal{R}_t^{i,u}$, and $\mathcal{P}_t^{i,u}$. The former is the set of all agents who chose agent $i$'s unchosen bandit at trial $t$, and got a reward (+1) at trial $t$, whereas the latter is the set of all agents who chose agent $i$'s unchosen bandit at trial $t$, and got a penalty (−1) at trial $t$.

It is easy to verify that, for all $i \in [1, n]$, for all $t \in [1, T]$, $\mathcal{R}_t^{i,c} \sqcup \mathcal{P}_t^{i,c} \sqcup \mathcal{R}_t^{i,u} \sqcup \mathcal{P}_t^{i,u} = [1, n]$.

For a given agent $i$, confirmatory information is information which confirms that agent $i$'s chosen bandit at trial $t$ was the "right" one (that is, the most rewarding one). Disconfirmatory information, on the other hand, suggests that agent $i$'s chosen bandit at trial $t$ was not the right one. Therefore, at trial $t$, agent $i$ considers the following information as confirmatory:

- A reward (+1) obtained by any agent $j$ who chose the same bandit as agent $i$—i.e., by any agent $j \in \mathcal{R}_t^{i,c}$;

- A penalty (−1) obtained by any agent $j$, $j \in [1, n]$, who chose the other bandit—i.e., by any agent $j \in \mathcal{P}_t^{i,u}$.

Similarly, at trial $t$, agent $i$ considers the following information as disconfirmatory:

- A penalty (−1) obtained by any agent $j$ who chose the same bandit as agent $i$—i.e., by any agent $j \in \mathcal{P}_t^{i,c}$;

- A reward (+1) obtained by any agent $j$ who chose the other bandit—i.e., by any agent $j \in \mathcal{R}_t^{i,u}$.

All agents update information they consider confirmatory with a learning rate $\alpha^+$, and information they consider disconfirmatory, with a learning rate $\alpha^-$. When $\alpha^+ > \alpha^-$, agents are said to have a confirmation bias. Similarly, one may define a "disconfirmation bias" when $\alpha^+ < \alpha^-$. The case $\alpha^+ = \alpha^-$ will be referred to as "unbiased".

**2. Update rule.** At every trial $t \in [1, T]$, all agents update the Q-value they associate with each bandit. In the next paragraphs, the following notation will be used:

- $Q_t^i(a_t^{i,c})$ denotes the Q-value associated by agent $i$, at trial $t$, to the bandit it chose at trial $t$;

- $Q_t^i(a_t^{i,u})$ denotes the Q-value associated by agent $i$, at trial $t$, to the other bandit at trial $t$.

CARL's modified Rescorla-Wagner rule can be expressed as follows. For any agent $i \in [1, n]$,

$$Q_{t+1}^i(a_t^{i,c}) = Q_t^i(a_t^{i,c}) + U_t^i(a_t^{i,c}) \tag{1}$$

$$Q_{t+1}^i(a_t^{i,u}) = Q_t^i(a_t^{i,u}) + U_t^i(a_t^{i,u}) \tag{2}$$

where

$$U_t^i(a_t^{i,c}) = \frac{1}{n}[\alpha^+ \sum_{j \in \mathcal{R}_t^{i,c}} (1 - Q_t^i(a_t^{i,c})) + \alpha^- \sum_{j \in \mathcal{P}_t^{i,c}} (-1 - Q_t^i(a_t^{i,c}))] \tag{3}$$

is agent $i$'s update of its chosen action $a_t^{i,c}$, and

$$U_t^i(a_t^{i,u}) = \frac{1}{n}[\alpha^+ \sum_{j \in \mathcal{P}_t^{i,u}} (-1 - Q_t^i(a_t^{i,u})) + \alpha^- \sum_{j \in \mathcal{R}_t^{i,u}} (1 - Q_t^i(a_t^{i,u}))] \tag{4}$$

is agent $i$'s update of its unchosen action $a_t^{i,u}$.

Therefore, for each agent and each bandit an update consists of the sum of prediction errors that were formed on the basis of all agents' payoffs. These payoffs are weighted differently—i.e., depending on whether the corresponding prediction errors are confirmatory or disconfirmatory. Each update sum is scaled by $\frac{1}{n}$, so that no update gets higher than 1, as in the original Rescorla-Wagner rule.

**3. Softmax policy.** At each trial $t$, each agent $i$ chooses an arm $a$ according to a softmax policy:

$$X_a^i(t) = \frac{1}{1 + exp(-\beta^i \times (Q_a^i(t) - Q_c^i(t)))} \tag{5}$$

where $X_a^i(t)$, called "policy" or "behavior profile", denotes the probability that agent $i$ chooses arm $a$ at $t$; $Q_a^i(t)$, the Q-value agent $i$ attributes to $a$ at $t$; $Q_c^i(t)$, the Q-value agent $i$ attributes to the other arm, $c$, at $t$; $\beta^i$, agent $i$'s inverse temperature, which quantifies agent $i$'s tendency to explore—i.e., to sample options it does not necessarily prefer. When $\beta^i = 0$, $i$ follows a random policy; when $\beta^i \to \infty$, $i$'s policy is completely determined by its preferences.

**4. Deterministic model.** In order to facilitate the analysis of CARL's learning dynamics, we derive its "deterministic equations" [27] in the two-agent case. To derive such a model, prediction-error updates are averaged over their probabilities of occurrence. As a consequence, Q-values' trajectories over time can be visualized in a single run, without simulation noise. Let $i$ and $j$ be two agents. We define $X_a^i(t)$ the probability that agent $i$ chooses action $a$ at trial $t$,

and $Q_t^i(a)$, the Q-value that agent $i$ attributes to action $a$ at time $t$. At time $t$, agent $i$ can update $Q_t^i(a)$ according to three different scenarios:

- agents $i$ and $j$ choose action $a$ at time $t$. This happens with probability $X_a^i(t) \cdot X_a^j(t)$.

- agent $i$ chooses action $a$ and agent $j$ chooses the other action at time $t$. This happens with probability $X_a^i(t).(1 - X_a^j(t))$.

- agent $j$ chooses action $a$ and agent $i$ chooses the other action at time $t$. This happens with probability $(1 - X_a^i(t)).X_a^j(t)$.

In the first two cases, agent $i$ updates positive prediction errors (which occur with probability $p_a$) with a higher learning rate $\alpha^+$. In the last case, action $a$ is the unchosen option so agent $i$ updates positive prediction errors with a lower learning rate $\alpha^-$. Thus, the expected update of option $a$ when it is chosen is:

$$\delta^c(t) = \alpha^+ p_a(1 - Q_t^i(a)) + \alpha^-(1 - p_a)(-1 - Q_t^i(a)) \tag{6}$$

And the expected update of option $a$ when it is unchosen is:

$$\delta^u(t) = \alpha^- p_a(1 - Q_t^i(a)) + \alpha^+(1 - p_a)(-1 - Q_t^i(a)) \tag{7}$$

To obtain the general expected update (that is, whether option $a$ is chosen or unchosen) of option $a$'s Q-value, one just needs to multiply $\delta^c(t)$ (*resp.*$\delta^u(t)$) by the probability that option $a$ is chosen (*resp.* unchosen) by agent $i$.

Therefore, at time $t + 1$, the expected Q-value update of action $a$ by agent $i$ is the following:

$$Q_{t+1}^i(a) = Q_t^i(a) + [2X_a^i(t) \cdot X_a^j(t) + X_a^i(t) \cdot (1 - X_a^j(t))] \cdot \delta^c(t) + (1 - X_t^i(a)X_a^i(t)) \cdot X_a^j(t) \cdot \delta^u(t) \tag{8}$$

Thus,

$$Q_{t+1}^i(a) = Q_t^i(a) + X_a^i(t) \cdot (1 + X_a^j(t)) \cdot \delta^c(t) + (1 - X_a^i(t)) \cdot X_a^j(t) \cdot \delta^u(t) \tag{9}$$

The behavior profiles $X_a^i(t)$ are obtained through the softmax formula.

## B. Simulations

Throughout the study, we investigated confirmation bias' effect on performance in various conditions using agent-based simulations. Within these, agents perform a static two-armed bandit task for a fixed number of trials. Agents find themselves in groups of different sizes, and can observe every other agent's actions and rewards—i.e., agents are modelled as nodes on a fully-connected graph. At each trial $t$, each agent $i$ chooses an arm $a$ according to a softmax policy (see Section IV A 3) with parameter $\beta$.

Throughout the study, all agents' inverse temperatures are set to $\beta = 4$. After agents make a choice, they receive a reward (+1) or a penalty (−1), which occurs according to the arms' reward probabilities. Following this payoff collection, agents update their respective Q-values according to the CARL model (see Section IV A 2). Afterwards, a new trial begins, and this sequence goes on until the final trial.

In most of our simulations, we investigated three types of environments, defined by the two options' reward probabilities. In a poor environment, obtaining an outcome of +1 is overall rare: the first arm's reward probability is $p_1 = 0.3$, and the second arm's, $p_2 = 0.1$. In a rich environment, the probability of getting a reward is overall high: $p_1 = 0.9$, and $p_2 = 0.7$. In a mixed environment, the probability to get a reward is overall neither low, nor high: $p_1 = 0.6$, and $p_2 = 0.4$.

We define performance as mean collected payoff per agent per trial (averaged over simulations). Performance improvement denotes the difference between actual performance, and expected performance when agents follow a random policy—i.e., their probability of choosing one or the other option is 0.5. This expected performance is: −0.6 in a poor environment; 0 in a mixed environment; 0.6 in a rich environment.

## Supporting information

**S1 Text. Effect of higher temperature (lower $\beta$).**
(PDF)

**S1 Fig. Mean collected payoff per trial for one agent in the group.**
(PDF)

**S2 Fig. Distributions of final Q-value gaps for one agent across 1000 simulations, as a function of bias strength.**
(PDF)

**S3 Fig. Mean final Q-value gaps for one agent across 1000 simulations, as a function of bias strength.**
(PDF)

**S4 Fig. Distributions of differences between agent 0's and other agents' Q-value gaps in a group of 5 agents, as a function of bias strength.**
(PDF)

**S5 Fig. Agents' Q-value gaps over time according to the deterministic model.**
(PDF)

**S6 Fig. Final polarization as a function of inverse temperature.**
(PDF)

**S7 Fig. Performance in all environments using different inverse temperatures $\beta$.**
(PDF)

**S8 Fig. Difference between confirmatory and unbiased agents' final Q-value gaps in various conditions.**
(PDF)

**S9 Fig. Evolution of Q-values over time in rich and poor environments, with $\beta = 1$.**
(PDF)

## Acknowledgments

Part of the abstract was generated with Writefull: https://x.writefull.com/abstract-generator.

## Author Contributions

**Conceptualization:** Clémence Bergerot, Wolfram Barfuss, Pawel Romanczuk.

**Data curation:** Clémence Bergerot.

**Formal analysis:** Clémence Bergerot, Wolfram Barfuss, Pawel Romanczuk.

**Investigation:** Clémence Bergerot.

**Methodology:** Clémence Bergerot, Wolfram Barfuss, Pawel Romanczuk.

**Project administration:** Pawel Romanczuk.

**Resources:** Wolfram Barfuss, Pawel Romanczuk.

**Software:** Clémence Bergerot.

**Supervision:** Wolfram Barfuss, Pawel Romanczuk.

**Validation:** Clémence Bergerot.

**Visualization:** Clémence Bergerot.

**Writing – original draft:** Clémence Bergerot.

**Writing – review & editing:** Clémence Bergerot, Wolfram Barfuss, Pawel Romanczuk.

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
