## [Decision Letter · Decision Letter 0]

30 Apr 2024

Dear Mme Bergerot, 

Thank you very much for submitting your manuscript "Moderate confirmation bias enhances collective decision-making in reinforcement-learning agents" for consideration at PLOS Computational Biology.

As with all papers reviewed by the journal, your manuscript was reviewed by members of the editorial board and by several independent reviewers. In light of the reviews (below this email), we would like to invite the resubmission of a significantly-revised version that takes into account the reviewers' comments.

We cannot make any decision about publication until we have seen the revised manuscript and your response to the reviewers' comments. Your revised manuscript is also likely to be sent to reviewers for further evaluation.

[1] A letter containing a detailed list of your responses to the review comments and a description of the changes you have made in the manuscript. Please note while forming your response, if your article is accepted, you may have the opportunity to make the peer review history publicly available. The record will include editor decision letters (with reviews) and your responses to reviewer comments. If eligible, we will contact you to opt in or out. Please refer to this https://osf.io/preprints/psyarxiv/kyfus to format your rebuttal letter.

Sincerely,

Stefano Palminteri

Academic Editor

PLOS Computational Biology

Daniele Marinazzo

Section Editor

PLOS Computational Biology

Reviewer's Responses to Questions

**Comments to the Authors:**

Reviewer #1: Review of Moderate confirmation bias enhances collective decision-making in reinforcement-learning agents by Bergerot et al.

Summary

In this study, the authors investigated the impact of confirmation bias on group performance in multi-agent simulations. To this end, they designed a new model that accounts for learning asymmetry in group reinforcement learning. The study simulates scenarios where agents observe others' actions and adjust their expectations based on the outcomes received by the group. The findings suggest that a moderate level of confirmation bias can enhance group performance in various cases, and this effect increases with group size. However, excessive bias strength or very small group sizes in a rich environment can lead to suboptimal outcomes.

The question investigated in the present study—What is the impact of confirmation bias on collective decision-making?—is intriguing, and various analyses as well as a novel computational model have been introduced to address it. The paper is clearly written, and the new model is very interesting. The analyses presented are relevant, and the figures are precise and informative. However, the analyses performed provide some insights into the question, but some mechanistic explanations of the observed effects are lacking. Furthermore, the collective aspect of the simulation appears limited, which in turn restricts the novelty of the results. Below, I introduce a series of major and minor comments. I believe that addressing the following issues would further enhance an already interesting study.

Major Comments

#1 “collective decision-making”

The authors discuss 'collective decision-making' from the title of the study. However, it appears that no collective decisions are actually made in the various simulations performed. Agents observe the actions of other group members and their associated outcomes, update their individual expectations, and make individual decisions. These individual performances are then aggregated and analysed. Could the authors explain the rationale behind using 'collective decision-making' to describe the decisions made in the simulations? From my understanding, the simulations presented in the paper lead to individual decisions that occur within a social environment and involve social learning.

#2 Mechanistic / computational explanation about the difference in resource scarcity.

The authors found that confirmation bias can be problematic in rich environments, especially in small groups. However, this result remains purely descriptive, and no mechanistic or computational explanation is provided for the origin of this effect. Could the authors provide more information about why this is the case?

#3 Explanation of why group size counteracts the effect of excessive confirmation bias

The authors also found that group size counteracts the negative effects of strong confirmation bias. However, no computational explanation is provided for why this is the case. Updating with social information appears equivalent to updating from multiple samplings of available options, with the number of samplings equaling the group size. The only difference being that the option preferred by the group would be more sampled. Consequently, it would not be surprising that an increase in group size enhances performance and reduces the negative impact of strong confirmation bias. With a larger group size, the (weighted) average outcome used to update individual expectations may become increasingly representative of the actual expected value of each bandit, limiting the likelihood that agents will select and adhere to the worst option. It would be interesting to analyze more closely the relationship between group size and the average outcome received, and how this relationship influences the positive effect of group size on mitigating strong confirmation bias. For instance, it would be beneficial to see a comparison of the average outcomes across groups of varying sizes, which presumably involves a decrease in the noise produced by binary outcomes. I believe this analysis is crucial for understanding the computations in the CARL model and the aforementioned group size effect.

#4 Unstable environment.

In the study, the authors manipulate the environment in one way, specifically through the scarcity of resources. Another common manipulation in a bandit setting is the (in)stability of the environment, where probabilities may reverse during the run. Surprisingly, environment stability remains unexplored in this study. On one hand, a biased agent would typically struggle to switch its decisions toward a newly optimal option in a changing environment. On the other hand, a group of agents biased towards both options (as is the case in some simulations presented here, where performance decreases) may potentially cope better with probability reversal. The diversity in strategies within these groups, which impedes performance in a stable environment, may counterintuitively enhance it in an unstable one.

#5 Agency and and confirmation bias in the CARL model

As mentioned in the manuscript, Chambon et al. found that agency is essential for activation of confirmation bias, and that observational learning alone does not appear to involve any confirmation bias in humans. In the current simulations, most of the information used by agents (depending on group size) is acquired through observation. One might then question whether the assumption made in the CARL model, that the level of bias is similar for one’s own outcomes versus those of other agents, is supported by the literature. I believe this is an important aspect of the study and should be addressed in the discussion.

Minor Comments

In the first Results section, the authors refer to the CARL model details in the Methods section. However, it would help the reader to gain insights about the update process directly in the Results section. Similarly, the softmax temperature is mentioned only in the Methods section. Considering its close connection to the values of learning rates, it would be beneficial for the reader to know its value for each simulation in the Results section.

Reviewer #2: Dear Editor,

Sorry for the delay. I would like to submit my review report for PCOMPBIOL-D-24-00479 entitled "Moderate confirmation bias enhance collective decision making in reinforcement learning agents" submitted to PloS Computational Biology.

Summary of the manuscript:

Using a multi-agent reinforcement learning model (Collective Asymmetric Reinforcement Learning), the authors investigated how "confirmation bias" in value-updating affects the performance of collective learning in a stationary two-armed bandit task. As discussed in the Introduction, behavioural biases found in the experience-based decision making (such as reinforcement learning) has been a hot topic in human decision sciences. Previous studies have identified conditions under which learning biases, such as confirmation bias, positivity bias, or "choice-confirmation bias" in value updating, can function adaptively for solo individual decision makers. However, how well such biases may perform in collective decision making have been under explored, even though humans often make decision under social influences. To fill this gap, the authors extend the RL model with two asymmetric learning rates (that is, one alpha for confirmatory update and the other for disconfirmatory update) to a fully-connected multi-agent setup where agents update their Q-values through both individual experience and social observations of others' payoff.

Systematically manipulating (1) the richness of the environment, (2) the strength on the choice-confirmation bias, and (3) group size, the authors show that the positive confirmation bias promotes collective intelligence under a wide range of assumptions, and that it performed especially well when group size was large. On the other hand, in the rich environment the bias had a detrimental effect on decision performance when group size was small. The smallest group size needed to overcome such a detrimental effect of confirmation bias became large with increasing strength of the bias. Such a group size effect was never observed in unbiased populations, and increasing group size could be harmful for disconfirmation biased population in the rich environment.

The mechanism behind their findings seemed to stem from the bifurcation in belief formation. When the choice-confirmation bias was strong, individual agents who happened to choose a suboptimal option gave a heavier weight to belief updating for reward of the chosen option than for reward information of unchosen optimal option, making the agent having a sort of "illusion" that the chosen option looks better. The richer the environment, the more likely it became that agents were misled by suboptimal rewards.

When there were other agents, however, the marginal effect of each payoff information became small because of the learning rate divided by N. In addition, the chance to learn about unchosen option's payoffs increased with increasing group size because there was an increased chance that someone chose the other option, making it easy to unlearn the illusion. Therefore, the cluster of suboptimal belief was hardly formed in large groups. This is my understanding how the group size effect emerges in the current model. On the other hand, when group size was too small, it could happen that all agents get stuck in a suboptimal option with having the illusionary belief. As a result, such a learnt illusion was more likely to emerge in a small group playing in the rich environment.

Evaluation:

The topic should be of wider interest in the field, and their modelling approach and results make sense. Very interesting work. I love it. I especially liked that the authors relied not only on agent-based stochastic simulations but used also the deterministic approach, which made their analysis more compelling. The manuscript is generally well-written.

However, I found a few major points that could fundamentally change how we should interpret the results. Below, please let me describe the major points, followed by some minor issues that I believe the authors should address before proceeding.

Major points:

(1-1) Test a wider range of beta, and connect it to the previous literature

The results reported seem to rely fully on inverse temperature (beta) being 4, which I believe is very high. In other words, agents assumed here were basically very exploitative (i.e. greedy). I guess their results would not change qualitatively even if they assumed a perfectly greedy choice policy (that is, beta = ∞). However, my huch is that their results would change dramatically under smaller beta values. The interpretation of the results seemed to depend heavily on this assumption.

The reason behind this prediction is that what they found here seems to be very similar to a phenomenon called the "hot stove effect" (Denrell 2007, March 1996). Although the classic hot stove effect assumed a symmetric (unbiased) learning rates, the mechanism is similar to the current study, that is, the long-lasting underestimation due to limited exploration. As I descried in the summary paragraph above, the group size effect came from the fact that agents in a group were able to access information from unchosen options, while solo individuals need to revisit the unfavourable option in order to learn it. Naturally, such limited information sampling is more pronounced when the choice policy is exploitative.

If the results were indeed explained well by such a biased sampling, the found effect of environmental richness, group size, and confirmation bias could be seen as one of possible ways to tweak the dynamics of information gathering. Then, the current work would become well connected to some of the previous literature including Denrell & Le Mens (2007; 2017); Toyokawa & Gaissmaier (2022).

It would be great if the authors can run additional analyses for other range of beta values and can provide an unified view, hopefully explaining a range of phenomena found in the current study as well as the previous finding.

Denrell J (2007) Adaptive learning and risk taking Psychological Review 114:177–187.

Denrell JLe Mens G (2007) Interdependent sampling and social influence Psychological Review 114:398–422.

Denrell JLe Mens G (2017) Information Sampling, Belief Synchronization, and Collective Illusions Management Science 63:528–547.

March JG (1996) Learning to be risk averse Psychological Review 103:309–319.

Toyokawa W, Gaissmaier W (2022) Conformist social learning leads to self-organised prevention against adverse bias in risky decision making eLife 11:e75308.

(1-2) What about other biases? How sensitive the current results are to the model's form?

Perhaps the author can think this is rather a future direction, but I wondered how their results might change by modifying the model's assumption. For instance, as mentioned in the Introduction, there are many different ways to implement asymmetry in learning rates, such as positivity bias (where alphas are implemented separately between positive and negative prediction error) or egocentric bias (alpha for self reward and alpha for social observation). Also, asymmetric alphas could act interactively with a choice inertia (Sugawara & Katahira, 2021), which could impact on the pattern found here. In particular, a positivity bias (and I think the choice-confirmation bias too) can also generate a choice inertia, and Sugawara & Katahira (2021) suggested that assuming a "choice trace" in softmax decision making would allow us to distinguish those two different mechanisms of choice inertia.

I think it would be fine for now to say such modifications could be put into the future direction, but it would be much appreciated if the authors can run some robustness tests using some minor tweaks in model assumptions. It is important to know which part of the results could be generalisable while other parts might be caused by a unique feature of the current model specification.

Sugawara, M., Katahira, K. Dissociation between asymmetric value updating and perseverance in human reinforcement learning. Sci Rep 11, 3574 (2021). https://doi.org/10.1038/s41598-020-80593-7

Minor points:

(2-1) Is "confirmation bias" an appropriate labelling?

I wondered if the labelling "confirmation bias" makes sense.

Yes, when beta was large like 4, agents' decision making was mostly greedy, choosing what they believed was the best. In such circumstances, it may make sense to assume that reward from the chosen option was a confirmatory cue supporting that what they chose was a good one. However, in lower beta cases, agents often choose options for exploration (choosing options with lower Q-values). I do not think that reward obtained from an exploratory choice necessarily "confirms" their original belief. In the Introduction, the authors mentioned the "choice-confirmation bias" which seemed to fit much better than just saying the confirmation bias. The authors should use "choice-confirmation bias" entirely in the main text.

Sincerely yours,

Reviewer #3: The present article discusses the impact of confirmation bias in a collective decision task, given different bias strengths and signs (confirmatory, disconfirmatory, or neutral) in various environments (rich, poor, mixed). The authors demonstrate that while adaptive in poor and mixed environments, confirmation bias can be detrimental in rich environments, especially when the population is small. Additional analysis shows the emergence of polarization within an interacting group in a rich environment when the agents present high confirmatory biases.

Overall, this work is highly relevant to social learning and uses appropriate methods.

A couple of aspects could be modified to enhance the clarity of the text and results. In particular, the methods section can be unnecessarily opaque at times, especially in the proofs. The following comments are ordered by priority level (from most major to most minor).

- 500 simulations were performed at the beginning, and this number is increased later on (probably to get clean distributions), but the number of trials also changes from the simulations in different environments, to the simulations with different biases, to the anaysis of final Q-value differences. The reasons for such a drastic augmentation in trial number (10-fold from the first to the last simulation !) are unclear.

- Why is the «performance improvement» defined in the methods not used? In the case of 1 agent, which acts as a control of the implementation of the learning algorithms, it seems that the mean payoff barely exceeds the expected performance upon random choice in Figure 1, even after removing the first 100 trials (panels d,e,f). Why is that? Was the learning completed?

- the computations presented in the methods need more detail. In particular, the proof presented in section 4 of the methods is difficult to understand. It should be at least mentioned that the way that the « deterministic » result is computed is an expectancy (which I am not entirely sure of) to get equation 8.

- line 247: why pick n=5 agents for this simulation and not more when the authors state from the previous simulation that the intersection point between strongly confirmatory and unbiased accuracy as a function of group size is between n=4 and n=6 (line 220)? Is it to detect the minimum group size to pass beyond the doubled-peak distribution of final Q-value gaps? If so, it should be mentioned in the text.

- Why choose to show the mean payoff instead of the average probability of picking the correct answer, which would be closer to the usual representation of a learning curve?

- In Figure 1, b is defined in the a,b, and c panels (referenced from section II A) when only introduced in section IIB in the text. I’d recommend defining it early on in section IIA.

- In Equation (5) of the methods, the * (convolution) should be replaced with an x (multiplication: \\times in LaTeX and Word) or with a dot, as in the rest of the equations.

- The notations are not always consistent throughout the text: the number of agents is sometimes denoted n (as in section II A) and sometimes N_agents (as in section II C).

-Similarly, the policy is sometimes referred to as X^i_a(t) (equation 5) and sometimes as X^i_t(a) (methods section 4). While it is understandable that these two notations are used in different contexts, they refer, in the end, to the same quantity. Using the same or different notations are two valid takes. Still, considering the journal's broad readership, and this research in particular being of potential interest to scientists from various fields, consistency might bring clarity to the paper.

- At the end of the methods, the reward probabilities associated with each environment could be written again to obtain a self-standing methods section and for better clarity.

**Have the authors made all data and (if applicable) computational code underlying the findings in their manuscript fully available?**

Reviewer #1: Yes

Reviewer #2: Yes

Reviewer #3: Yes

PLOS authors have the option to publish the peer review history of their article (what does this mean?). If published, this will include your full peer review and any attached files.

Reviewer #1: No

Reviewer #2: **Yes: **Wataru Toyokawa

Reviewer #3: No
---

## [Decision Letter · Decision Letter 1]

9 Aug 2024

Dear Ms. Bergerot,

We are pleased to inform you that your manuscript 'Moderate confirmation bias enhances decision-making in groups of reinforcement-learning agents' has been provisionally accepted for publication in PLOS Computational Biology.

Best regards,

Stefano Palminteri

Academic Editor

PLOS Computational Biology

Daniele Marinazzo

Section Editor

PLOS Computational Biology

Reviewer's Responses to Questions

**Comments to the Authors:**

Reviewer #1: I thank the authors for their detailed responses to my concerns. They provided pertinent analyses and additional discussion points that substantially improve the study.

Reviewer #2: Dear Editor,

I apologize for the delayed response. The authors have thoroughly addressed all the reviewers' comments point-by-point, resulting in substantial improvements to the revised manuscript. Their interpretations of the results in relation to the known "hot stove effect" are both compelling and interesting. Additionally, the authors have provided a fair and well-written discussion on the limitations of their study. I have no further concerns for the authors to address.

Congratulations to the authors on their excellent work.

All the best,

Wataru Toyokawa

Reviewer #3: Thank you for making the amends and answering my questions. I think that the content is now clearer, and I am happy to recommend this paper for acceptance.

**Have the authors made all data and (if applicable) computational code underlying the findings in their manuscript fully available?**

Reviewer #1: Yes

Reviewer #2: Yes

Reviewer #3: Yes

PLOS authors have the option to publish the peer review history of their article (what does this mean?). If published, this will include your full peer review and any attached files.

Reviewer #1: No

Reviewer #2: **Yes: **Wataru Toyokawa

Reviewer #3: No

---

## [Editor Report · Acceptance letter]

28 Aug 2024

PCOMPBIOL-D-24-00479R1 

Moderate confirmation bias enhances decision-making in groups of reinforcement-learning agents

Dear Dr Bergerot,

I am pleased to inform you that your manuscript has been formally accepted for publication in PLOS Computational Biology. Your manuscript is now with our production department and you will be notified of the publication date in due course.

With kind regards,

Anita Estes
